# Exploring Behavioral Anthropomorphism With Robots in Virtual Reality

Chinmay Wadgaonkar, Johannes Freischuetz, Akshaya Agrawal, Heather Knight
wadgaonc,jofreisc,agrawaak,knighth@oregonstate.edu
Collaborative Robots and Intelligent Systems (CoRIS)
Oregon State University
Corvallis, Oregon, USA

## ABSTRACT

Virtual reality (VR) and social robotics have mutual benefits. VR offers an instrumented and manipulable environment in which robots and people can virtually interact as well as tools for visual manipulations of robot materiality and color. VR also has a wealth of knowledge about how multimodal communications like motion, proxemics, and touch can inform interaction. Submersing social robots in VR provides an opportunity for physically-grounded interaction that leverages behavioral anthropomorphism. This work attempts to intersect these previously disparate areas, eliciting participant storytelling about the simplest possible anthropomorphizable robot: a robot that approaches and then bumps into you. In the study, 16 participants experience twelve manifestations of virtual/physical robots that approach and collide into them. The moment of collision provides an opportunity for expressive interpretation that offers a first glimpse into future potentials for physically embodied companion characters in virtual reality.

## CCS CONCEPTS

• **Computer systems organization** → **Robotics**; • **Computing methodologies** → **Mixed / augmented reality**; **Virtual reality**; • **Human-centered computing** → *User studies*.

## KEYWORDS

Robot Behavioral Anthropomorphism, Robots in Virtual Reality, Robot Materiality, Social Robotics

**ACM Reference Format:**
Chinmay Wadgaonkar, Johannes Freischuetz, Akshaya Agrawal, Heather Knight. 2021. Exploring Behavioral Anthropomorphism With Robots in Virtual Reality. In *Proceedings of Boulder '21 (Virtual, Augmented, and Mixed Reality for HRI)*. ACM, New York, NY, USA, 8 pages. https://doi.org/10.1145/nnnnnnn.nnnnnnn

## 1 INTRODUCTION

Recent developments in social robotics have suggested the idea of *behavioral anthropomorphism*, in which even robots with very simple forms can be perceived as having human characteristics,

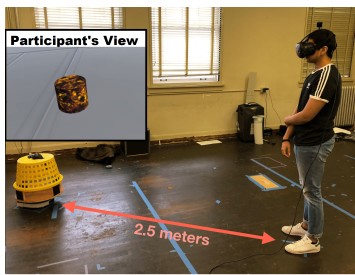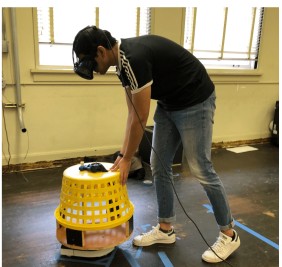

(a) Before Approach      (b) After Approach

**Figure 1: This study consisted of an evaluation of a physical robot synced with 12 different apparent materials (lava pictured). Participants watched the virtual version of this robot as it approached and bumped into them. The apparent material had a large impact on participant feelings of safety and robot aggressiveness ratings.**

for example, when they move in a way that expresses they have a particular goal or intention[21]. Researchers are gaining insights into how simple robots forms can communicate via motion, light, and sound in this fascinating but growing domain[11][1]. These developing understandings of simple robot communicative capabilities have been shown to enable people to anticipate and understand current robot operation state, and can help cue desired behavior from the human, but there is much left to learn. For example, diverse exploration of the impact of visual form would require the building of many robots, which limits the work to date. To address this gap, the opportunity explored in this paper is ways in which explorations of robot form and material VR might bootstrap the development of new understandings in this area.

Why now? Gaming in on-screen and virtual environments is a huge and growing industry that has significantly reduced the cost-of-entry for developing 3D virtual worlds. For example, the Unity game engine offers stores where one can purchase or download, for free, a variety of pre-developed environments, textures, objects, and even characters, at a low cost. This provides a unique opportunity to further explore questions of the impact of visual features of robotics on the ways in which people attribute particular social characteristics, goals, intentions, and communications. This is novel relative to previous efforts [19], as we are emphasizing the sociability of this varied form, and the impact of the visual features of attribution of anthropomorphic characteristics to our simple robot. Our efforts complement ongoing efforts to integrate haptics and physical embodiment in VR, such as hardware the user can lay in, to experience being a flying bird while wind blows in their

face (a fan). This has been shown to increase the believably of VR environments because sensory experiences are shown to increase a sense of place "place illusion", and the plausibility to the player that the scene is actually happening "plausibility illusion" [23].

Combining these two potentials, this work leverages the presence of a literal robot and the flexibility of VR visual experience to further the understandings of the way in which materiality might impact human interpretations of robot (or VR-character) behavioral anthropomorphism. Further understandings of behavioral anthropomorphism would enable robots to be incorporated, smoothly and effectively, in natural human environments, and communicate where necessary, e.g., gesturing for someone to move out of the way during navigation and/or inviting them to come take a seat on a robot chair, approaches that are less interruptive and often more socially acceptable than just verbally commanding a human to move out of the way. This can also be supported by visuals that vary the apparent characteristics of the physical robot in virtual reality. In this paper, we present a study on how humans perceive a robot's behavior when a minimal real-world robot is augmented in virtual reality, allowing humans to interact with the robot in both the real and virtual world. As an early work in this space, we also seek to identify whether and how mixed reality spaces might allow more futuristic, adaptable robot design explorations, that also leverage the libraries of existing Unity materials in this increasingly accessible VR backdrop.

Specifically, the presented user study consists of two simple robot shapes that approach and collide with a person wearing a VR headset. The motion of the real physical robot is mapped to a VIVE VR hand-controller that was mounted on the robot to leverage this new virtual environment. To clarify the reality of the physical robot, we further leverage our own previous work on communicatory collisions [18], asking: *What happens when a physical/virtual entertainment robot collides with someone in a mixed physical/virtual space from a behavioral anthropomorphism perspective?* The research conditions we select leverage previous research results. For example, Sirkin et. al[21] describe a mechanical ottoman that people thought communicated like a pet in shaking to get someone to offer it their feet, thus inspiring simple forms such as a cylinder or a cube. Further, Zlotowski et al[27] have discussed the potential benefits and challenges that may arise by building anthropomorphic robots, which leads into which design conditions impact perception. The shape of a robot and the path a robot takes play an important role of how people anthropomorphize a robot, but the most important condition of a robot is its materiality.

The second goal of this study was to collect utterance examples of ways in which apparent characteristics of the robot in VR impacted behavioral anthropomorphism (Section 6), for example, ways in which participants attribute character, intent, and storytelling to the robots across experimental conditions. To gain data here, we asked our 16 participants to describe what happened, the kind of character the object is, their relationship, and their mutual reactions and attitudes ("think out loud protocol") .

Our results suggest that there is high research and productization value for utilizing VR contexts to further our understanding of robot behavioral anthropomorphism in a mixed reality space. From the participants' perspective, this construct allowed them to imagine a future with adaptable, flexible, highly integrative robot forms.

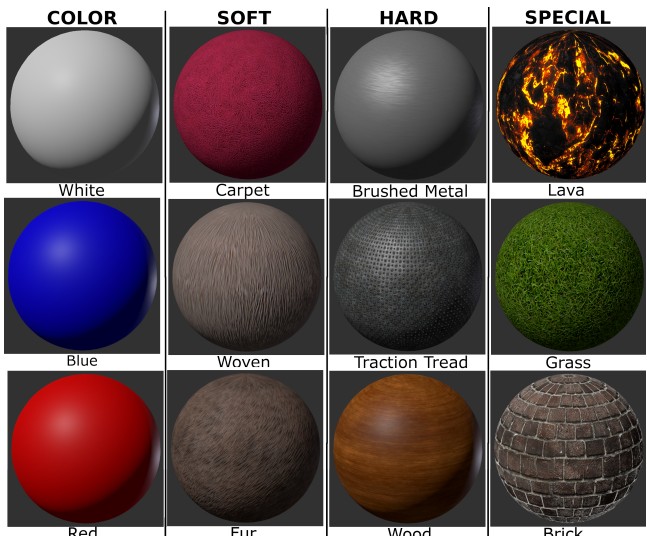

**Figure 2: The 12 VR Materials: three base colors, three soft materials, three hard materials, and three special materials, which were all available for free from the Unity store.**

It should also be mentioned that the net cost of the VR software resources used to develop this experiment were zero. The material libraries, forms, and environments we used were all available to download for free, underscoring the accessibility of continuing explorations, particularly as this technology continues to improve.

## 2 RELATED WORK

### 2.1 Prior work in social robotics and behavioral anthropomorphism

It is well known throughout the HRI community that humans tend to anthropomorphize the behaviors of robots [27]. This holds true even for minimal designs, such as furniture [1, 9, 10]. Moving with intent is a key feature of this attribution of animacy and intent [5]. For example, acceleration can change the apparent emotion and level of energy [16]. The question of this work is whether motion intersects with form to changes the metaphors the people may use to interpret robot intentions. Related prior work investigating robot form include [20], in which multi-legged robots were perceived as having higher aggression when compared to wheeled robots. Physical features such as shape [8] and height [15] have additionally been shown to influence people's attributions and expectations of dominance in robots. The question of this work is whether materiality can have a similarly strong effect. After all, attributions impact interpretation [12]. Similarly, predefined associations around colors or materials might leak into our perceptions of minimal robots.

### 2.2 Robot Motion and Communicatory Collisions

Robots communicating through active touch, a.k.a., *collision based communication* is relatively new – [18] and [1] – but demonstrates that collisions can change people's sense of a robot's intention

and persona. Such collisions are literally the end point of a robot's expressive motion pathway, combining prior work in robot motion expression and social touch. Prior work in socially-informed robot motion (e.g., velocity and attention to goal) is diverse [11] [13] [6] [22] [14], and impacts human likelihood of interacting and fluently sharing space with robots. Additional work in robot social touch demonstrates ways in which robots can imitate similar human inspired communications [26], [25], [2]. These provide foundations and inspirations for the present work.

## 2.3 Applications for physical objects and robots in virtual spaces

Finally, haptics has been investigated for use in VR in previous work, but the social analysis of these objects is less common. Most current VR touch applications (haptics) seek to increase perception of place [7]. Socially-grounded examples do demonstrate the benefits of haptics in VR for perceived co-presence in virtual teams[17]. This physical touch need not be represented by a human-like avatar to evoke behavioral anthropomorphism. Simeone et al[19] investigated the influence of artificially created mismatch between the virtual object and its physical counterpart on a human's perception of objects. By substituting physical objects with slightly discrepant virtual representations, they found that the materiality of objects can significantly impact the believability of a substitution [19]. In addition, Tachi et al. [24] have developed a Shape Approximation Device, a robot whose end effector is capable of simulating contact with a variety of surfaces such as those having vertices, edges, and concave or convex parts. The uniqueness of this paper is in including both the physical and the social.

## 3 MIXED REALITY TECHNOLOGY DESIGN

The system is set up in multiple parts that all interact together to give the participant the experience of seeming to interact with several unique robots with different appearances, individually, while only actually interacting with one physical robot. This is achieved by having (1) a virtual representation of the robot that can take on various apparent shapes and materials, (2) physical toppers that mirror the shape of the virtual robot, and (3) physical robots that can be controlled with a PS4 controller.

## 3.1 Virtual Robot: Tracking & Calibration

The virtual representation of the robot tracks the real robot as it changes position and rotation. If the real robot changes rotation in any axis the virtual robot will match this, allowing the person to touch the real robot tracked to the virtual robot. This tracking is done by using the VR controller positioned on top of the real robot in a holder as shown in Fig. 4.

The tracking system needs to be calibrated before each trial. This is done by marking 4 points equidistant from each other on the real-world object with the controller. The tracking system then saves these 4 points. The controller is then placed in the holder shown in Fig. 4 which was 3D printed using [3]. Once the controller is successfully mounted to the object in real life, the button can be pressed again to let the tracking system use the 4 points to find the offset for the position and rotation of the object. This process can be seen in Fig. 4. This tracking offset can be used for the cube and

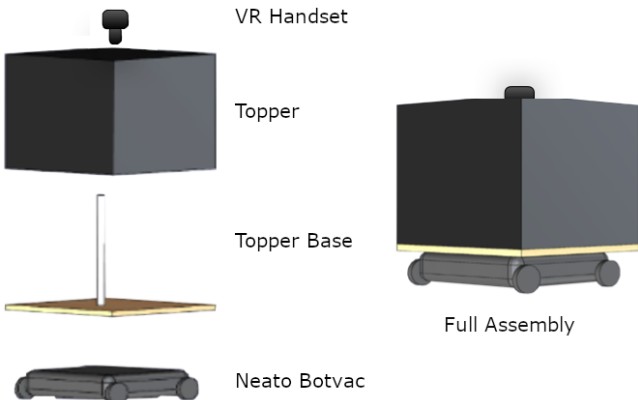

**Figure 3: Breakout view of the VR Prototyping System Components: Robot Motion Via Neato Botvac, Topper Base allows for easy switch out of Robot Topper Shapes (cube displayed here), and a VIVE VR Hand Controller is mounted at top center for best tracking from all angles.**

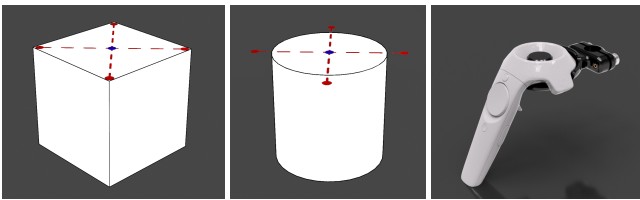

**Figure 4: Topper Shapes and Calibration: This project utilized a freely available Solidworks model [3] that we 3D printed to hold the HTC Vive Controller. A hand controller was used to click four points (red dots) around the periphery of the physical object. The controller was then placed in the holder in the center of the object (blue dot). This calibrated the virtual robot to the size and location of the real robot.**

cylinder and does not need to be calculated during the transitions in the study as the math from the previous calibration can be used [4].

## 3.2 Physical Robot & Toppers

The robot in this experiment is a modified Neato robot with an attached topper to allow the robot to have the shape of a cube or a cylinder (Fig. 3). This allows the participant to touch the topper matched to the virtual robot. The topper is attached using 4 bolts that are connected through the chassis. This topper is a square piece of plywood that sits slightly larger than the robot itself so that the participant only interacts with the topper and does not accidentally hit the actual robot. This piece of wood then has a hollow cardboard cube connected to the top of it to give the shape of the topper without adding unnecessary weight. Inside of this cardboard cube, sits a pipe that goes up to the top of the cube and has a 3D printed holder for the controller. This allows the controller to be held securely allowing for accurate tracking.

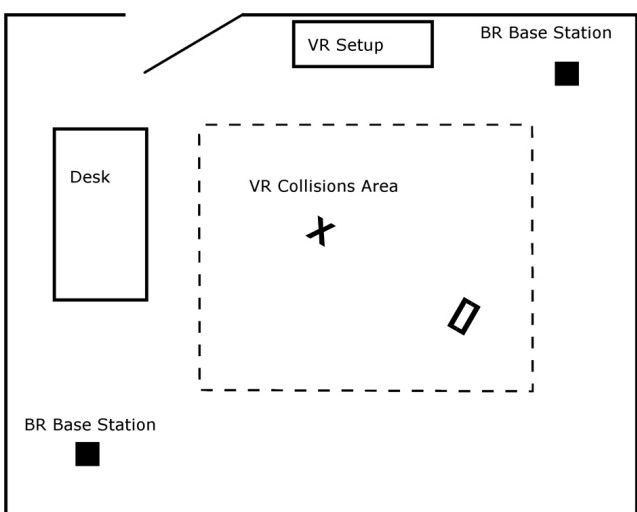

**Figure 5: Study Room Layout included the Vive Base Stations for tracking, and desks for the study conductor and technology operator. The participant stood on the X on the floor, while the robot began each trial in a taped out square.**

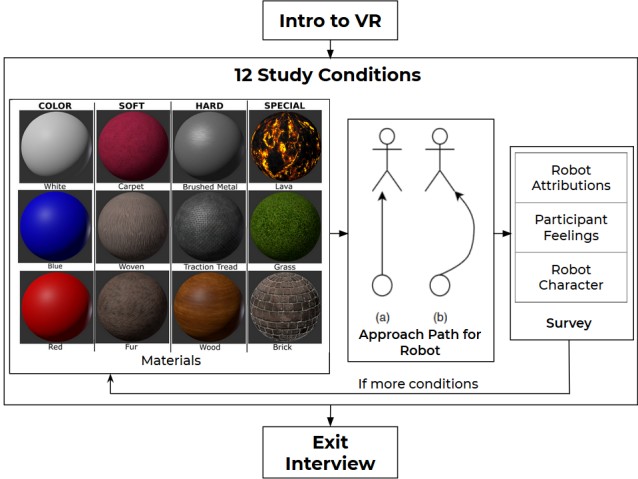

**Figure 6: During each trial setup, a virtual material was layered onto a model of the current robot shape while the participant was turned away. The robot conducted its approach and collision behavior, and finally, the participant responded orally to a six question Likert scale survey.**

Another modification was accessing the control system to allow for custom remote control. There is a port inside of the Neato that can be easily accessed by opening the top compartment that allows for control of the movement. A Raspberry Pi was used to act as a relay to connect a laptop to the Neato wirelessly. Since the Raspberry Pi needs power, a portable battery is included in the Neato's compartment. A PlayStation controller is connected to the laptop and a program interprets and sends messages based on button presses to the raspberry pi relay allowing for remote control via a PlayStation controller over Wi-Fi.

## 4 EXPERIMENT DESIGN AND METHODS

In this section we discuss the variety of materials used for developing the apparent robot (in VR), different shapes of the robot and the two paths taken by the robot to approach the participants. We also discuss the detailed procedure for the experiment followed by the survey and interview structure as also illustrated in Fig. 6.

### 4.1 Material, Shape, and Path Manipulations

The research manipulations spanned apparent material (VR only), topper shape (VR + physical matched), and robot approach path (VR + physical matched).

*Material*: The apparent material of the robot was our most important manipulation, as it showed how easily robot physical appearance can be changed in VR. We included 12 different materials depicted in Fig. 2. These materials were intended to explore soft and hard materials, but also included flat colors as a baseline, and special VR colors as an exploration of the VR-material space.

*Shape*: The robot as seen by the participant in VR was counterbalanced between two shapes, namely *cylinder* and *cube* as seen in Fig 4. The shape of the physical robot was also changed to the same shape as the one in VR so that the participant could see and feel

the same structure of the robot. These shapes were chosen due to quick build time in order to validate the method. The research team also believed these shapes to be easier to associate to different real time objects that the participants may encounter on a daily basis.

*Approach Path*: The robot approaching the participant took two different paths that was counterbalanced across the trials (Fig. 10). These two paths result in two types of collision. The first path is *direct*, where the robot approaches the participant in a straight line and results on a direct collection with the participant along the path of approach. The second path is *indirect*, where the robot approaches the participant in an arc and collides indirectly to the side of the participant (Fig. 6).

We pre-recorded the robot's motions in different Robot Operating System (ROS) Bags and then mapped the PS4 controller's buttons to trigger a specific ROS Bag to ensure consistency. This ensured that the robot had constant motion for each participant and reduced any chances of human errors when moving the robots. A hard-coded speed was used to keep the motion consistent among participants. Battery level did sometimes decrease velocity as the charge went down, thus we ran studies in batches of two and charged the battery between trials.

### 4.2 Procedure

The study was conducted in an experiment room on a university campus over multiple days (Fig. 5). There was one study conductor and one technology operator. The study conductor guided the participant through the trials and asked the participant survey questions for data collection in addition to controlling the robot's motion. The technology operator controlled the virtual reality. One external camera was used to record the interaction between the robot and the participant.

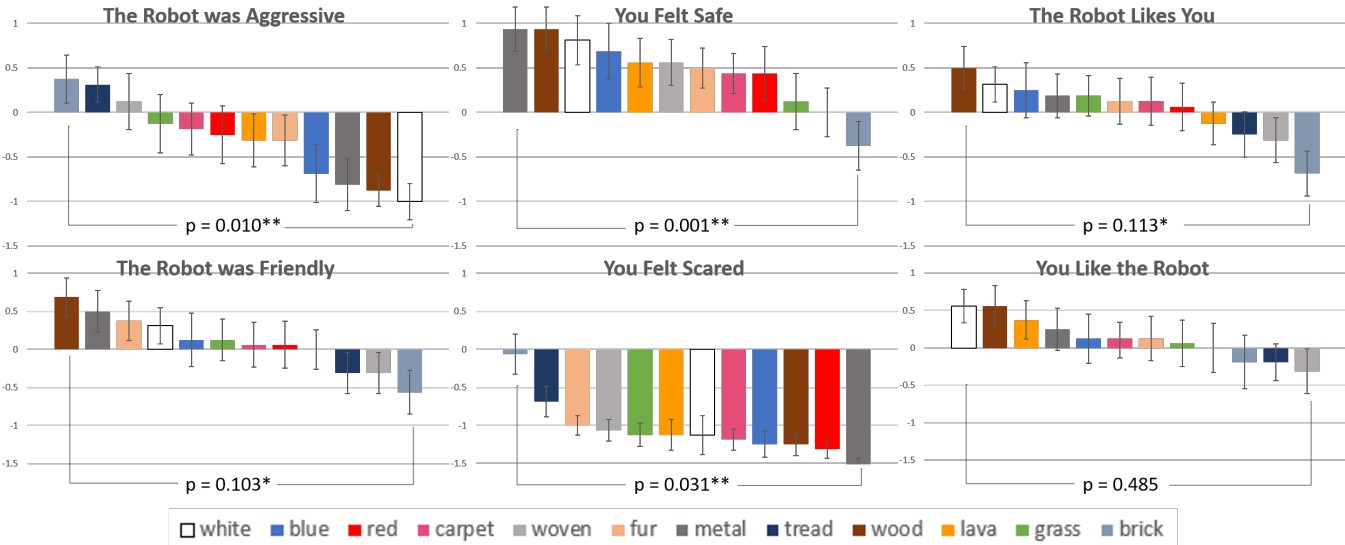

**Figure 7: Material Means for the Six Survey Categories**

After a quick introduction to VR, the participant was put into the VR headset. First, they oriented themselves to the virtual environment, an abstract building structure that is empty except for the robot (Fig. 1). Next, the participant was introduced to the robot, asked to touch the robot and see if the appearance of the robot in the VR maps to what they feel in the physical world. This step was used to calibrate the robot (as per Sec. 3). Next, they were guided to stand on the X (also marked on the floor in the real room) and the robot went to its starting position (Fig. 5).

Each trial began with the participant facing the robot. Fig. 1 shows what the room and the robot looked like to the participant. The robot then approached the participant in a direct or indirect path Fig. (6) and collided with the participant. After the collision, the robot would retreat a couple inches and stop. The participant was asked survey questions, which they responded to on a 5-point Likert scale, and open ended follow-up questions in which the participant would often anthropomorphize the robot. These questions are described in Sec. 4.3. The technology operator transcribed these participant responses live. The robot would then move back to its starting position. The participant would turn away from the robot toward the study conductor at the front desk, and that would conclude the trial.

Six such trials were conducted with one robot shape, while varying the texture and the path of approach. There was a short break before the next six paths in which the participant stayed in VR (facing away from the robot), while the physical topper of the robot was switched to another shape and re-calibrated. This was followed by 6 more trials with varying textures and paths of approach. Between the trials, the participant was asked to face the desk allowing us to avoid sudden changes of material or shape being visible to the participant (Fig. 5). After the completion of all 12 trials, the participant was asked several new open ended questions.

### 4.3 Participant Surveys and Interviews

Throughout the experiment, participants were encouraged to comment on what was happening and their impressions of the robot. After each trial, the study conductor specifically asked them to report on their impressions of the trial, what they thought the robot was trying to do, what the robot's attitude was toward them, and anything else they imagined or contemplated during the trial. We also prompted them to comment on the appearance and behaviors of the robot if had not already mentioned them.

These post-trial open-ended questions were followed by six survey questions: (1) *The robot was friendly.* (2) *The robot was aggressive.* (3) *You felt scared.* (4) *You felt safe.* (5) *The robot liked you.* (6) *You liked the robot.* Participants responded verbally using five-point Likert scales from 'Strongly Agree" to 'Strongly Disagree." This question format was introduced before entering the VR environment. They were also invited to comment on their answers to these questions after the fact, and the conductor occasionally asked them to explain ratings if they had high variance from earlier trials or made a rating that seemed contradictory to an earlier comment.

At the end of the experiment, participants anthropomorphized the robot when they were asked to comment on their experience and features of the robot. We asked participants specifically to comment on their most and least favorite robot, material, shape, path, and share their observations about the impact of each research manipulation.

## 5 SURVEY RESULTS: ROBOT ATTRIBUTIONS, EXPERIENCE, AND RELATIONSHIP

The final dataset included 16 participants, each of whom participated in 12 trials, resulting in 192 unique datapoints. Robot material was a significant predictor of almost all participant ratings (Fig. 7) supporting the power of robot 'costuming' in influencing human perceptions of robot character and intent. Material did not have a significant impact on ratings of the statement, "you like the robot,"

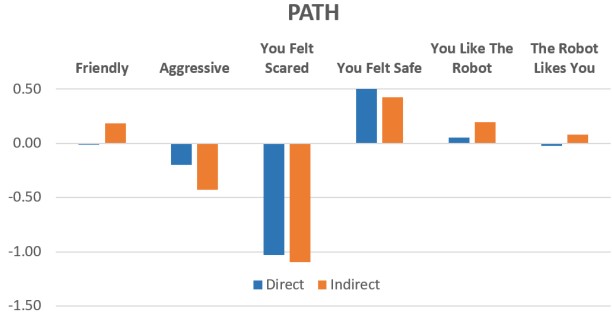

**Figure 8: Approach Path: Means for the Six Surveys**

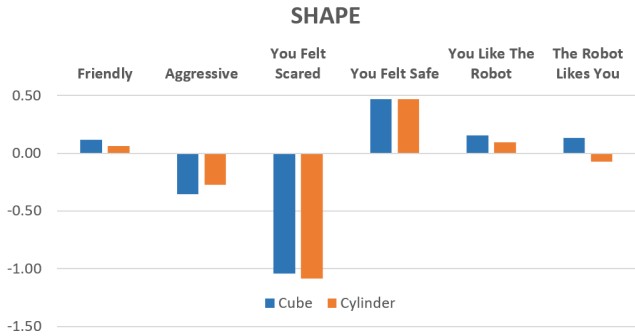

**Figure 9: Robot Shape: Means for the Six Surveys**

indicating that participants have similar perceptions of the robots but have different preferences as individuals. Surprisingly, approach path and robot shape did not significantly impact participant ratings (Fig. 8 and Fig. 9).

Statistical significance is calculated with the non-parametric Kruskal-Wallis H test. The Kruskal-Wallis H test is used to determine if there are any statistically significant differences between two or more groups of an independent variable on a continuous or ordinal dependent variable. The three independent variables used here were Material, Path and Shape. This test is considered the non-parametric alternative to the one-way ANOVA and an extension of the Mann-Whitney U test, a test that allows the comparison of two independent groups. All significant differences found using the Kruskal-Wallis test were also found using Multi-Factor ANOVA. Both tests achieved similar statistical significance results, validating our results. The five-point Likert scale ratings for the independent variables, as described in section 4, resulted in the ordinal nature of the dependent variables.

We organize our results below into the attribution results of whether people found the robot to be friendly/aggressive, the comfort results of whether people felt scared/safe, and the camaraderie results of whether they liked the robot, or thought the robot liked them (this section), followed by examples of robot anthropomorphism.

## 5.1 Friendly/Aggressive Results

As can be seen in Fig. 7, the first two Likert questions we asked participants were whether the robot was friendly and whether the robot was aggressive. A Kruskal-Wallis H test showed that while robot material was a significant predictor of robot aggression ratings ($\chi^2$=24.819, p=0.01), it did not statistically significantly predict robot friendliness ($\chi^2$=17.159, p=0.103).

Numerically, the friendliest robot materials included wood, metal ('gray'), fur and white. If we remember the way participants interpreted metal as a gray that matched the environment in the previous sub-section, this may explain why this robot seems friendly because it fit in so well. Coordinating robots to their environmental context is an interesting idea for future work. The metal texture was also rated as one of the least aggressive, just above white, and wood.

If wood is considered a not aggressive and friendly material, perhaps more roboticists should consider integrating wood into real world robot designs. The organic texture was highly recognizable in VR, beating out even the best categorized soft material (carpet) which was rated only neutrally friendly and moderately not aggressive. The white-colored robot also bodes well for many of the companion robot designs currently popular, as its mean was the least aggressive of all. Numerically, an indirect path was rated less aggressive than a direct path (Fig. 8).

## 5.2 Scared/Safe Results

As can be seen in Fig. 7, two Likert questions we asked participants after the collision were whether they felt scared and whether they felt safe. A Kruskal-Wallis H test showed that there was a statistically significant difference in both scared and safe ratings between the 12 different materials. For feeling scared, $\chi^2$=32.035, p=0.001. For feeling safe, $\chi^2$=21.228, p=0.031. Material significantly predicted survey ratings of scared and safe. Numerically, metal ('gray'), red and wood resulted in the lowest survey ratings for feeling scared and metal ('gray'), wood and white resulted in the highest survey ratings for feeling safe. Alluding to the comments made in the previous section about the metal ('gray') material, participants may have have found a material that matched the environment to be more safe and less scary.

The brick material as per Fig. 7 numerically ranked the lowest in the participant's feeling safe as getting hit by a pile of bricks was described to be intimidating. Additionally, brick was also ranked the highest in the materials that made people feel scared.

## 5.3 Participant Liking Robot/Robot Liking Participant Results

As can be seen in Fig. 7, the last two Likert questions we asked participants after the collision were whether they liked the robot and whether they thought the robot liked them. A Kruskal-Wallis H test showed that there was not a statistically significant difference in neither of these two questions' ratings between the 12 different materials. For liking the robot, $\chi^2$=10.509, p=0.485. For feeling safe, $\chi^2$=16.837, p=0.113. Material did not predict survey ratings of scared and safe. Numerically, an indirect path was rated more likable than a direct path. Against our hypotheses, shape had no impact on these two survey ratings (Fig. 9).

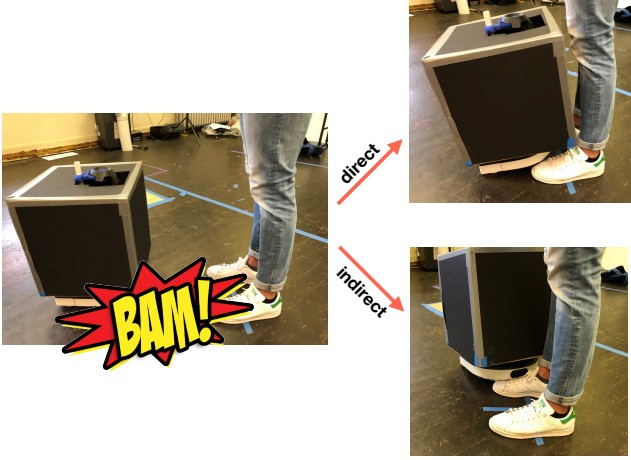

**Figure 10: Participants came up with diverse explanations of robot motives and characteristics, varying with apparent material and path, as shown in the qualitative results.**

## 6 QUALITATIVE RESULTS: PARTICIPANT STORYTELLING AND REACTIONS

This section offers exemplar quotes illustrating some of the story-telling behind participant ratings, namely: path, collision, materiality, and participant response.

The **robot motion pathway** played a significant role in the participants descriptions of about robot intent and relationship to them, despite the lack of significant quantitative results. For example, an example of a statement in which the robot was described as friendly: "It felt like a pet coming over to me and sniffing me out, like a silly robot in a movie, and a pet, gave it a quirky feeling." Which contrasts to one in which it was perceived as aggressive: "The fact that its camo and moved that way suggests evasive maneuvers. It's a guerilla warfare robot, a Vietnam vet who was trying to make me think he was going past me. It could also have been trying to snipe me, get me from behind. "Such pathways also impacted interpretations of robot intent, for example, curiosity: "It reminded me of a situation where you're trying to sneak up on a cat from the side and pet it. It was clearly interested in me, trying to get close to me without triggering any alarm." Finally, we mention cases that illustrate the negative experience of robots that get it wrong on human experience: "This one doesn't give a ****. It is not aggressive per se, friendly-ish but comes off as confused or a little bit drunk. Meandering somewhere, like the one on its phone earlier. It wasn't expecting to run into me." There were also several instances of people thinking its sensors were broken, rather than seeing expressivity: "It's clearly trying to go somewhere or I am in the wrong place and it wants me to go there," or that they were the one in error, "Instead of saying "Sorry, can I get by?", it's saying "Please move now!"" These examples illustrate the vast potentials for future work seeking to further understand the impact of materiality in combination with relative motion behaviors and robot non-verbal communications.

Next, we consider the impact of **collision** on storytelling, with a particular focus on strong reactions or entertaining interpretations,

as those inform some of the extremes in the survey data. For example, "My first thought was that it was an overly aggressive drunk girl in a bar… I was surprised it bumped into me and backed up. I thought it might stop. Maybe it was trying to get my attention." Again, a small signal can be magnified into complex interpretations with a few subtle context cues. We also heard concepts involving collaboration or co-working with the robots, e.g., "We were both doing something and I was going to move out of the way but we bumped into each other" deepening the anthropomorphic story-telling, "we laughed, my coworker the brick robot who occasionally changes texture and flashes." But of course, collision is not always seen as a friendly action: "It didn't let go when it got my foot. This one seems a lot more threatening and aggressive." And lastly, it was also common for people to judge the robot as incapable after a collision: "This robot seemed really dumb. It was not aware that I was there. Very indirect path, delayed reaction on a bump, not a very smart robot this time. It was trying to move from A to B and didn't know I was there at all."

Finally, **apparent robot materiality** often led to participant descriptions of robot job and character. For example, "the white color reminds me of... a *hospital*... It makes me imagine pressing on it to reveal something like in an *operating room*" or "This is a green food delivery robot. It should deliver [locally-sourced food restaurant] or something." or alternatively to the grass or wood robots, "It reminds me of nature. It feels like it is right out of nature." In addition to jobs, the color often provoked associations of likeness or in-groupness, e.g.,"It's an ottoman, a pink one. This one wants to do my nails. It has to be a girl because it's pink." Or another person decorating his house said, "I would totally put that in my house on a zebra-skinned rug." For unusual textures, anthropomorphism became more detailed: "It looks like lava… This apparently came from hell with a lot of aggressiveness, hostility," or "I assumed it would bump straight into me since red means danger." Finally, the more industrial materials sometimes lead to literal interpretation: "This is almost like a structure that you put up to prevent cars from going into a parking area or landscape element. I feel like he's on a construction job, he's going to carry some things over." The descriptions here could be leveraged further in both robotics and VR character design, as costuming/materiality is an unexplored area in robotics that appears to have high storytelling impact.

Outside of the experimental conditions, a subset of participants also physically interacted towards the robots, trying new **movement responses** themselves as the relatively long (12 conditions) experiment continued, like, moving out of the way, asking to sit on the robot, letting the robot go through their legs, and one even jumping over it. This shows the deep potential for physical interaction and play in this mixed reality format.

## 7 CONCLUSION

This paper suggests several rich potentials for leveraging robot anthropomorphism in virtual reality. Adapting insights from social robotics into the VR setting, it suggests that there is a utility to exploring variants of robot appearance and physicality because of the low barriers to entry and the many software assets that already exist for free. We present the software that we have developed (and are happy to share) that localizes a simple robot into a VIVE VR

system and overviews our calibration techniques and user interface. Using this system, the main study used evaluates participant anthropomorphization across twelve apparent robot materials (Fig. 2, and two shapes. Each participant experienced all materials across one of two path conditions, acting as an impulse to social interpretations, which all ended by bumping into the human, to clarify the physical presence and also offer a point for interpretation.

As expected, anthropomorphization of the behaviors was by and large the most common participant response. Beyond this overarching observation, our results illustrate both expected and unexpected social results. For example, color theory did not necessarily predict people's attributions to the robots, with the red robots ranked as the second-least scary. This highlights the importance of running experiments like this to discover such variants and may indicate that concepts like shade are probably important. Further, the ability of the system to render a material could support or confound an expected result, with soft materials often hitting the limits of current VR resolution systems, but recognizable hard objects like brinks rated as highly unsafe when barreling toward you. People are visual creatures with an estimated 2/3 of our brain real estate dedicated to visual processing, thus, leveraging the ability of virtual environments is an efficient way to explore factors related to robot physical experience. These materiality results illustrated the power of VR in rapid, flexible, low-cost explorations of character attributions for social robots.

Finally, the openness of some of participants to using motion to communicate back with the robot – combined with the breadth of character-based storytelling – underscores the varied potentials for future work that incorporates bi-directional interactivity, social sequence, and even narrative into an extended set of interactions between robot and human in mixed-reality. Such future implementations/experiments, may offer further capability to do VR prototypes for social robots in the real world, and/or as a novel entertainment options to deepen character-based interactions in mixed-reality.

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
