# OpenReview forum: "Exploring Behavioral Anthropomorphism With Robots in Virtual Reality"
_humanrobotinteraction.org/HRI/2021I/Workshop/VAM-HRI — VAM-HRI 2021 Oral_

### Official Review · AnonReviewer3 · 2021-03-04
**Easy to read, investigates an interesting and novel problem, is well motivated, and offers an interesting approach**

**Rating:** 8
**Confidence:** 5

**Review:**

This paper investigates behavioral anthropompishm for robots using a virtual reality interface. This work calibrates a VR representation of a robot with the pose of a real robot with the same shape to change the visual experience of the user based on the perceived material. This is an interesting approach because it offers a minimal way to augment real-world robots with virtual reality to potentially impact how people attribute social characteristic, goals and intentions to robots. This paper conducted a study where a real robot approach and collided a person wearing a VR headset (who could see a virtual robot located at the pose of the real robot), and the robot shape, material and approach path was augmented in VR. 16 users participants were involved in this study, and users filled out surveys on various impressions such what they felt happened, their relationship, and mutual reactions and attitudes to the robot. The results in this paper are very interesting and well-detailed, discussing the important implications on using VR to better understand robot behavioral anthropomorishm in mixed reality environments. The section on how users generated stories to justify what they saw based on the robot’s material and approach path was particularly interesting and is useful for future researchers interested in designing both real and virtual robots.

Overall, this paper is easy to read, investigates an interesting and novel problem, is well motivated, and offers an interesting approach to studying behavioral anthropomorphism for robots in mixed reality contexts. The user-study was also well designed and the analysis reveals insights into how users perceive robot’s based on form and action. Overall, i recommend this paper be accepted.

 Comments and questions to investigate:
- The authors should make explicitly clear whether the study was within-subjects or between subjects (it seems like not all subjects experienced the same shape-texture combinations?)
- there is a missing paranthese in conclusion “Using this system, the main study used evaluates participant anthropomorphization across twelve apparent robot materials (Fig. 2, and two shapes”
- It would be interesting to see how these results integrate with more knowledge of the robots motion intent, for example, what if the robot’s intended path was visualized?  (for example, see the paper: "Communicating and controlling robot arm motion intent through mixed-reality head-mounted displays”.)
- How important is it that the real robot and the virtual robot are correctly calibrated? In practice in unstructured environments there is decalibration, and would be interesting to see what the effects of this are.
- What if the virtual robot shape has a mismatch between the real robot shape?

---

### Official Review · AnonReviewer2 · 2021-03-08
**Would love to see this in the context of calibrating expectations proximal human-robot interactions**

**Rating:** 7
**Confidence:** 4

**Review:**

Interesting paper. I liked the framing of the problem in terms of how people assign intentions to the robot, and its study through the VR setup. Though in my mind, I think there is a killer application of this sort of setup in calibrating expectations in proximal interactions between humans and robots. E.g. how space is demarcated on a factory floor (based on the robot's internals) may be different from what expectations the human workers are attaching to the robot and that may impact teamwork. One might try to use virtual overlays to affect expectations of the user to the correct level. Here is an example: https://www.colorado.edu/today/2020/10/21/pufferfish-inspired-robot-could-improve-drone-safety

A couple of questions on the approach:

1. How do you know that the impact time is exactly calibrated to the VR world? I would imagine a slight difference there would completely alter the results. For example, if the robot bumps into the subject unexpectedly before it does in VR.
2. Since the setup involves the same person reacting on multiple trials, I think an analysis of the statistics of responses in each trial slot is necessary. I don't think that the number of participants is high enough to make independent calculations by randomizing the order of trials.

---

### Decision · Program_Chairs · 2021-03-06

Accept (Oral)